# On-the-Fly Fusion of Remotely-Sensed Big Data Using an Elastic Computing Paradigm with a Containerized Spark Engine on Kubernetes

**DOI:** 10.3390/s21092971

**Published:** 2021-04-23

**Authors:** Wei Huang, Jianzhong Zhou, Dongying Zhang

**Affiliations:** School of Civil and Hydraulic Engineering, Huazhong University of Science and Technology, Wuhan 430074, China; huangwei0316@hust.edu.cn (W.H.); zhangdongying@hust.edu.cn (D.Z.)

**Keywords:** fusion algorithm, parallel computing, big data, Kubernetes, cloud computing

## Abstract

Remotely-sensed satellite image fusion is indispensable for the generation of long-term gap-free Earth observation data. While cloud computing (CC) provides the big picture for RS big data (RSBD), the fundamental question of the efficient fusion of RSBD on CC platforms has not yet been settled. To this end, we propose a lightweight cloud-native framework for the elastic processing of RSBD in this study. With the scaling mechanisms provided by both the Infrastructure as a Service (IaaS) and Platform as a Services (PaaS) of CC, the Spark-on-Kubernetes operator model running in the framework can enhance the efficiency of Spark-based algorithms without considering bottlenecks such as task latency caused by an unbalanced workload, and can ease the burden to tune the performance parameters for their parallel algorithms. Internally, we propose a task scheduling mechanism (TSM) to dynamically change the Spark executor pods’ affinities to the computing hosts. The TSM learns the workload of a computing host. Learning from the ratio between the number of completed and failed tasks on a computing host, the TSM dispatches Spark executor pods to newer and less-overwhelmed computing hosts. In order to illustrate the advantage, we implement a parallel enhanced spatial and temporal adaptive reflectance fusion model (PESTARFM) to enable the efficient fusion of big RS images with a Spark aggregation function. We construct an OpenStack cloud computing environment to test the usability of the framework. According to the experiments, TSM can improve the performance of the PESTARFM using only PaaS scaling to about 11.7%. When using both the IaaS and PaaS scaling, the maximum performance gain with the TSM can be even greater than 13.6%. The fusion of such big Sentinel and PlanetScope images requires less than 4 min in the experimental environment.

## 1. Introduction

Cloud computing (CC) has shown its strength for bootstrapping genomic data analysis in life science, and has unquestionably influenced earth science [1,2]. Cloud-based RS big data (RSBD) production systems (CCRSBDPS) have become prevalent in recent years [3]. CC has shown significant potential in massive RS data storage and processing, on-demand services, and information services in domains such as drought monitoring, ecology assessment, and crop yield prediction. The ability to generate time-series RS data archives with an on-demand parallel has created vast opportunities for advanced natural resource monitoring and process understanding. Because no sensor provides satellite images with high spatial resolution and high temporal resolution, the development of RS fusion algorithms (RSFAs) is indispensable to produce time-series observations of the Earth’s surface structure and content [4]. While RSFAs play a vital role in data mining in CCRSBDPS, few works direct an efficient fusion of heterogenous RSBD on CC platforms.

To this end, many of the efforts made to speed up processing systems for RSBD are from the information technology community (ITC) and geoscience community. Most researchers from ITC focus on scaling RS algorithms with big data processing frameworks (BDPF) in diversified cloud deployments (public clouds, private clouds, hybrid clouds, and community clouds). For example, Sun et al. proposed an optimized framework for RSBD processing in CC [5]. They introduced a quantum-inspired evolutionary algorithm for the Spark engine to optimize the task scheduling in the proposed framework. The Spark engine is the de-facto BDPF in CC, and is widely employed in the geoscience community to classify and detect the changes in a large volume of multispectral RS images and synthetic aperture radar images [6]. Many works focus the optimization of task scheduling for the Spark engine [7]. However, due to the inherent heterogeneity of CC platforms, task scheduling is still considered an NP-hard problem [8]. Because different cloud providers may adopt different CPU and network virtualization technologies [9], the modified Spark engine may have a huge performance difference. As evidence, practical research found that Spark-based algorithms using Docker containers have a performance advantage over the algorithms using virtual machines (VMs) [10,11]. Containerization is a lightweight and novel virtualization technology. Traditional hypervisor-based virtualization systems such as Xen, VMware, and KVM provide multi-tenancy and hardware independence for a guest operating system (OS) in the form of VMs. Containerization systems such as the Docker engine, OpenVZ, and LXC using application virtualization provide similar independence in containers. Because application containers managed by the Docker engine share the same OS kernel, the number of containers on a computing device is much higher than the number of VMs [10]. Containerization provides an opportunity to enhance the scalability of BDPFs. Accelerating BDPFs using lightweight virtualization frameworks on enterprise clouds is promising. However, Spark does not allow for the runtime scalability of the resources associated with its executors, which means that the Spark computing resources are invariable once they are allocated to a Spark application. To bridge the gap, researchers add elasticity to the Spark engine. For example, a recent study proposed a dynaSpark, with which the number of allocated CPU cores used by Spark executors in a Docker container can be adaptively changed [12]. While Spark’s performance is not linear with increasing resource allocation for containers, containerized Spark in a cloud environment has completely changed life science. The development and deployment of time-critical cloud-native applications using containerization have prevailed in recent years [13].

Other than the above methods, more and more researchers from the geoscience community are adopting the Google Earth Engine (GEE) cloud platform to target the volume challenge of RSBD. GEE is a public cloud platform specialized for planetary-scale geospatial analysis, which improves the analysis of massive RS data that was not feasible using a desktop processing machine. The surging publications have testified to the rapid adoption of GEE [14]. Recently, a study introduced a highly scalable temporal adaptive reflectance fusion model algorithm (HISTARFM) to use the elastic computing resources of GEE [15]. While HISTARFM shows an enhanced performance advantage over ESTARFM, the parallel implementation of HISTARFM is hidden. An unprecedented volume of RS data stored in other cloud platforms needs efficient fusion algorithms. Generally, there is a large number of RS image fusion algorithms (RSIFAs). However, based on Dian’s experiments, most of the fusion algorithms suffered from high computational complexities because they depend on iteratively to solve the complex optimization problem [16]. Graphic process units (GPU) are commonly used for accelerating image processing. Because GPUs are massively parallel processors with high processing power and memory bandwidth, GPUs can process data multiple times faster than CPUs. GPU-accelerated image processing libraries such as CLIJ are widely used [17]. Deep learning-based spatiotemporal fusion methods can thus be significantly accelerated via the powerful computational abilities of GPUs [18]. Recently, Hong proposed a novel extension of Spark, which offers GPU-accelerated scalable computing [19]. While these libraries and Spark extensions show significant potential to improve the scalability of RSIFAs, there is little research focusing on the parallel RSIFAs that take advantage of the scalable computing paradigm.

In this study, we are the first to introduce the K8s operator model that explores the elastic computing paradigm in cloud platforms. We propose a framework that integrates K8s with containerized Spark, with consideration of their advantage and disadvantages. K8s is the standard for deploying containerized applications at scale in a multi-cloud environment. K8s operators can be considered to be a control loop for K8s applications to obtain customizable resources without an advanced declaration. Starting from version 2.4.0, Spark can run on K8s clusters in the client model. However, the inconvenience is that manual configurations are required in order to build networking channels between Spark diver (SD) hosts and Spark executor containers (SEC). Spark critical performance parameters should be manually predefined, which is a restriction to the use of elastic computing resources. The memory space and instance number of SEC are unchangeable once the algorithms are submitted. Our framework requires less parameter configuration. The framework only requires the hints on the memory allocated to the SD, and can update the resource configuration of SEC. Because the containerized Spark can be booted in a few seconds [20], the number of cores and the memory allocated to SEC can be dynamically changed to adapt to the workload. We exploit the merits of collocated containers on the same computing host while avoiding overloading the host. Because Spark executor exchanges task results with each other for fault tolerance, the inter-container communication latency of containers on the same host would be negligible. In order to constrain the performance degradation caused by the resource competition of these containers [21], we propose a task scheduling mechanism (TSM) to change the containers’ affinities with the hosts based on a weighted average policy. The TSM adjusts the pod affinities to a host after referencing the ratio between the number of completed and failed tasks at the runtime. In order to illustrate the advantages, we construct an OpenStack cloud computing environment. The scaling methods provided by both the IaaS and PaaS layer are used by the TSM. We implement a parallel enhanced spatial and temporal adaptive reflectance fusion model (PESRARFM) using a Spark user-defined aggregation function (UDAF) to reduce the number of joining operations. The PESRARFM requires less than 4 min to fuse five RS big images (about 10 GB) with the proposed framework, while the traditional ESTARFM takes dozens of hours to complete.

The remainder of this paper is as follows. Section 2 describes the experimental datasets, and then the cloud-based processing framework is introduced and followed by the TSM and PESRARFM. Section 3 describes the experimental cloud computing environment and explains the scaling methods used by the framework. Section 4 provides an in-depth discussion of the performance of the framework. We summarize the results from the previous chapters in Section 5, and provide the novelties of our research and future directions in Section 6.

## 2. Data and Method

### 2.1. Experimental Data

As shown in Table 1, Sentinel-2 is an earth observation mission from the European Union Copernicus Programme that systematically acquires optical imagery at high spatial resolutions (10 m, 20 m, and 60 m) over land and coastal waters. The mission is a constellation with two twin satellites (Sentinel-2A and Sentinel-2B), and with a revisit of 5 days. The temporal resolution may be unsuitable for daily monitoring applications such as precision agriculture. Commercial satellite corporations such as Planet provide daily coverage Earth observations. PlanetScope is one of the satellite constellations operated by Planet. The complete PlanetScope constellation consists of approximately 130 satellites, and it enables the daily monitoring of the entire land surface of the earth (equating to a daily collection capacity of 340 million km^2^/day). While the spatial and temporal resolutions are both high, the images are expensive. Therefore, a precision agriculture application would require the fusion of these images. State-of-art engines adopt HDFS to store unprecedented volume of RS images [22]. Spark-based parallel programming models can be used to process massive RS images [23]. 

### 2.2. The Proposed Cloud-Based Processing Framework

In this section, we propose a cloud-based processing framework. As illustrated in Figure 1, the proposed lightweight framework consists of the cloud orchestration layer, the Service-Oriented Computing Layer, and the cloud portal. The cloud portal is responsible for resource management, execution state monitoring, supporting user authentication, and accounting.

The cloud resource layer is the infrastructure as a service layer. This layer consists of storage, computing, and networking resources. The computing resources are virtual machine instances managed by hypervisors, bare-metal machines (BM), and cloud providers such as Alibaba, Amazon, and Google Cloud. The top layer is a unified cloud platform portal which supports user authentication, service accounting, and image and configuration management. The user authentication service is responsible for multi-tenancy access to cloud computing resources. The service accounting is responsible for security, pricing, and other aspects. The image management service manages image registration on the cloud platforms. The configuration management service supports all service configurations, such as defining monitoring strategies and updating resource quotas of applications.

In the service-oriented computing layer, we adopt a container-based microservices architecture for easily updating the lossely-coupled service components. Lightweight working components include RS file storage services, the RS object storage service, RS key-value storage services, the Prometheus service, the network function virtualization service, and K8s API services. K8s API is an open-source toolkit that manages the container-based applications in an effective, automated, and scalable manner (https://github.com/operator-framework (accessed on 23 March 2021)). These long-running storage services rely on network virtualization for communication with each other. Typically, network virtualization functions have different features. For instance, Weave Net creates a virtual network that connects containers across multiple hosts and enables their automatic discovery [24]. Calico provides a highly scalable network with high throughput rates that would be valuable for data-intensive parallel algorithms [25]. Note that the performance comparison of the virtual networks is outside the scope of this study. The workflow of the proposed framework is as follows.

First, authenticated cloud users submit computing jobs to the platforms, then the application manager (AM) provides application management services for the users. After receiving the application specifications from the cloud portal layer, the AM registers the submissions as application objects (AO) in order to manage their life cycle. Application specifications are programmable with Yet Another Markup Language (YAML) files, which specify the types of applications, the container images used to run the applications, the restart policy when a failure occurs, and the specification of storage volumes and computing resources. The AM manages application dependency, retrying failed applications, updating application resource requirements, and scheduling task executions based on predefined policies. Once the application object has been built and the resource requirements have been mapped to the custom resources definitions (CRDs), the AM will create an application controller (AC) for the AO. The AC uses a working queue (WQ) to manage the instances of AOs. The WQ watches all types of events related to these AOs. Once created, the WQ records all of the events of the instance until its completion. In the framework, the Prometheus service would collect the failed or successful events, and would send the status and failed reasons to the WQ. The AC continually checks whether the K8s cluster can match the CRD, and uses a reconciling mechanism to request the K8s to create, update and delete AOs if the K8s cannot match the CRDs. For convenience, we call the procedure as the ‘K8s operator model’.

We implement the application management service based on the K8s operator model. The Spark AM accepts the submissions of the user’s Spark applications. Then, Spark AM creates a Spark controller for the construction of a K8s namespace used by the Spark application. The Spark controller is a long-living micro-service. Once added to the working queue of the Spark controller, the Spark controller will create a Spark operator for the Spark application. The Spark controller manages the lifecycle of the Spark operator. The Spark operator watches the pods of the Spark driver and the executors of the application instance. We extend the original Spark operator to collect metrics such as the running counts, the failed counts, and the total number of pods. These metrics will be exports to the Prometheus monitoring service. The Spark controller can use the metrics to schedule the tasks of Spark applications.

#### 2.2.1. Proposed Task-Scheduling Mechanism

In order to improve the efficiency of Spark applications on K8s clusters, we propose a task scheduling mechanism (TSM) in this work. The key idea is to dynamically change the task affinities of the Spark application with the computing nodes. This is the only practical solution for Spark on the K8s model. We use the mutating admission webhook (MAW) of K8s to control the Spark pod assignments. MAW intercepts the resources request of the Spark pods and updates the pod specifications before these pods are scheduled to the computing node. The TSM considers the following constraints and conditions.

Each computing node has a region label. Typically, the network communication cost in the same region is smaller than the inter-region communication. The Spark executor replicates the blocks of resilient distributed datasets and transmits them to other Spark executors for fault tolerance. Thus, the TSM preferably assigns Spark executor pods to the nodes in the same region. As we know, running too many pods on the same node would cause intense competition among the pods. Therefore, our scheduling algorithm avoids assigning many pods to the same node in order to mitigate the unbalanced workload. Currently, there are many workload prediction algorithms in the cloud. However, the adaptive workload prediction systems are dependent on sophisticated system architecture. Users may have no right to access these systems. Normally, the ratio of the failed counts of the task pods to the successful counts of task pods is a good indicator to reflect the workload characteristics. The newly added computing resources are typically less busy. As mentioned earlier, the Spark AM can update the resource requirement specification, and the Spark controller can therefore change the task pod assignment if this information is used in this context.

Based on the above condition, the TSM for the Spark controller should operate under the constraints of the following equations. Suppose there are H hosts in R regions, where each host has several cores, and memory space. Core*_ij_* means that h_i_ has j cores, and Mem*_ij_* means that host *h*_i_ has *j* megabytes of memory. Assume that the cores and memory requirements of the Spark driver pods (SDP) of the Spark applications are not changeable, and are defined by users. Spark application objects (SAOs) are added to the working queue (WQ), and the WQ will assign the priority (PRI) for each SAO. The spark executor pods (SEPs) execute parallel tasks during their life cycle. Because the parallel tasks are similar in the transformation chain of Spark applications, we assume that each pod executes only one task for discussion convenience. Here, the Core and Mem represent the number of cores and the amount of memory storage used by each SEP, respectively. AF*_ij_* represents the degree of affinities of the SEP*_i_* with the host h*_j_*. AF*_ij_* = 0 means that the *i*-th pod should not run on the h*_j_*. AF*_ij_* = 1 means that the *j*-th pod should run on the h_j_. SEP has an age attribute in terms of the total running time of the pod in a host. The constraints are formally expressed as follows.

We use CP and FP to represent the number of successes and failures of the executed tasks of a SEP, respectively. Once a SEP has finished a batch of tasks successfully, it would continue processing the next batch. Given a period *µ*, the probability of the number of completed SEPs satisfies the Poisson distribution. The larger the probability is, the more stable the host is. Here, the *x* is the actual number of completed pods; *e* is approximately equal to 2.718. The goal is to minimize the time cost, given the dynamic nature of the SEPs.
$$Total\_core=\overset{n}{\underset{i=1}{\sum }}core_{ij}\;j\geq 1$$
$$Total\_mem=\sum_{i=1}^{n}mem_{ij}\;j\geq 2048$$
$$p\left(x;\mu \right)=\left(e^{-\mu }\right)\left(\mu^{x}\right)/x!$$

TSM works as follows. First, TSM intercepts the resource specification of Spark applications. Push them to the WQ and initialize a default priority for each SAO. Then, the TSM checks the total CPU core requirement of the SDPs in order to decide whether to request new computing resources from the cloud platforms. The region of an SDP determines the assignment of SEPs of the same SAO. For each SAO, TSM assigns the SEPs to that region with sufficient memory storage. If the memory storage is insufficient, the TSM assigns the SEPs to other regions with small networking communication latency with the SDP. Otherwise, it would traverse the hosts of the region and choose the least recently used computing host. In the reconciling stage, the TSM dequeues the SAO that has the highest priority. Typically, the number of SEP pods is large, while the number of computing hosts is small. Therefore, dynamically changing the affinities of the SEP pods with the computing hosts is beneficial for the improvement of the efficiency of the SAO. The TSM employs a user-friendly strategy to satisfy their expectation. As shown in Figure 2, the input of the TSM includes a set of Spark algorithms (S), a working queue (wq), and a given watching period μ. In line 18, the users can express the scheduling expectation on whether or not to place the tasks on the newer node, on a node which is less overloaded, or randomly by specifying the relevant weights. For instance, a user may expect to select a computing host with a high probability to complete more tasks by setting the weight (W_2_) to a value range from 0 to 1. The TSM will rank the affinities and generate the scheduling policy in order to assign each SEP to a computing host. The TSM also dynamically changes the priority of each SAO in the WQ; if the failure rate of an SAO is too large, it will be rescheduled with a higher priority. Finally, the TSM will notify K8s to update the resource requirement and wait a short period.

#### 2.2.2. Proposed Parallel Fusion Algorithm

Now that the framework has been built for Spark applications on cloud platforms with the scheduling mechanism, another question is how to design efficient parallel fusion algorithms for big RS images. Currently, most Spark-based RS algorithms are implemented with low-level resilient distributed datasets (RDDs) [23]. Considering that an algorithm requires the changing of the value of a pixel in a big image, the direct implementation based on the strip-oriented programming model is filtering all of the irrelevant rows from multiple RDDs, which will waste computing resources. Another scene is image convolution operations that require pixels that may traverse different rows. A developer has to join multiple RDDs in order to construct the needed data block for the pixels, which forces the Spark engine to shuffle multiple RDDs partitions, and thus requires unconstrained data transmission.

In order to solve the problem, we incorporate the Spark DataFrame abstraction for the mapping of RS big images to the resilient distributed tables. The Spark Dataframe supports the window operation on the big tables with user-defined aggregation functions (UDAF) [26]. The window represents a collection of rows of RS images, which has a unique row key. The Spark SQL conducts a query against the big tables. For example, the widely used normalized difference vegetation index (NDVI) can use the SQL statement “select (NIR-RED)/(NIR+RED) as NDVI from RS_IMAGE”. The ‘select’ relational operator is a projection operation that takes the specified red (RED) and near-infrared (NIR) band columns from the big table. The arithmetic examples (+, −) are operators of the SQL expression objects. More specifically, the Spark Catalyst of the Spark engine will analyze the Spark SQL statement and generate optimized physical planning in order to apply processing logic to the RS big images. Mapping the query statements to processing logic goes through the following three stages: an analysis stage, a logical optimization stage, and a physical stage. In the analysis stage, an abstract syntax tree is constructed in order to generate the logic plan. In the logical optimization stage, standard optimization operations—such as projection pruning, null propagation, and other rules—are applied to the syntax tree. In the physical stage, relational operators are mapped to low-level parallel primitives of the Spark engine. The Spark Catalyst engine will generate computing tasks that access the pixels in the relevant window. Rather than accessing the whole RS images for the image convolution operations, the engine can significantly reduce the computing costs. For convenience, we call the method the ‘windows-oriented programming model’ (WPM).

We present the parallel enhanced spatial and temporal adaptive reflectance fusion model (PESTARFM) based on the above model by analyzing the procedure of the ESTRAFM and comparing the novities of the model to the strip-oriented programming model (SPM). ESTARFM is the most widely-used fusion algorithm that assumes the spectral reflectance difference of the RS images derived from different satellite sensors capturing the same area in the same day can be modeled by linear functions [27]. Specifically, given a pair of pure, homogeneous coarse-resolution and fine-resolution pixels from an RS image, the spectral reflectance difference of the pair of pixels is only caused by systematic error. ESTARFM assumes that the spectral reflectance of a mixed pixel can be modeled as a linear combination of the reflectance of the different land cover components in the pixel. ESTARFM assumes that the change of reflectance of each land cover component is linear during a short period, and that the proportion of each component is stable. Therefore, ESTARFM treats the change of the spectral reflectance of the coarse-resolution images during the short period as the linear combination of the changes of the spectral reflectance of each land cover type of the fine-resolution images during the period. Consequently, once the change rate of a land cover type is obtained during the short period, ESTARFM estimates the unknown spectral reflectance of the pixels of the land cover type according to the change rate and the change of the reflectance of the pixels of the coarse-resolution images during the short period. In order to accurately predict the unknown spectral reflectance of a pixel at time T_p_, the algorithm searches ‘similar pixels’ (pixels within the same area as the central pixel) with the same land cover in the window from the coarse-resolution image at time T_0_, and leverages the reflectance changes of all of the similar pixels within the window of the coarse-resolution image from times T_0_ and T_p_. As shown in Figure 3. the flowchart of ESTARFM is as follows.

First, the searching of similar neighbor pixels (SSP) is conducted. The input of the algorithm consists of the following five scenes of RS images: the fine-resolution images at times tm and tn, and the coarse-resolution images at tm, tn, and tp. The output of the algorithm is the estimated value for each pixel of the fine-resolution image at tp. In order to distinguish the pixels of the input images, the algorithm uses the center pixel abstraction to represent the pixel in the output image. The SSP identifies the similar pixels for each pixel within a predefined window of the fine-resolution images at times tm and tn, respectively. The output of the SSP is the intersection of the two results to obtain an accurate set of similar pixels. Supposing that we are searching similar pixels within a 5 × 5 window from the two fine-resolution images, the implementation of PESTARFM using SPM requires fives filtering and mapping operations and four joining operations. Second, ESTARFM needs to predict the reflectance changes by calculating the weighted contribution of similar pixels. The contribution of each similar pixel is weighted by the location of the pixel and the spectral similarity between the fine- and coarse-resolution pixels. Intuitively, the higher similarity and smaller distance of the similar pixel to the central pixel has a higher weight. In order to obtain a good representation of the spectral similarity, ESTARFM combines the spectral reflectance of the coarse and the fine resolution images of the two different dates and calculates the correlation coefficient between each similar pixel and its corresponding coarse-resolution pixel of the two images. Therefore, the second stage will require three joining operations to obtain the reflectance changes. In the third stage, ESTARFM calculates the conversion coefficient in order to determine the changing rate for each similar pixel in a search window. This stage is rather similar to the second stage, which also requires three joining operations with SPM. Finally, in order to calculate the reflectance for each center pixel, the final stage requires two joining operations with the SPM. The first operation is to calculate the temporal weight, and the second operation is to calculate the final reflectance from the two estimations according to the temporal weights.

While PESTARFM based on the SPM requires nine joining operations, the PESTARFM based on the WPM requires only four joining and one window operation. The PESTARFM source code based on the WPM refers to (https://github.com/huangwei2913/RsImageFusion (accessed on 23 March 2021)). First, the four joining operations map the five images to a distributed table; second, the window operation calculates the standard deviations of the images at T_m_ and T_n_. Typically, this stage is quite fast. The deviations can be easily broadcast to the distributed table. We implemented a “RSImageFusionUDAF” class to manage the window operation. The processing chain that includes searching similar pixels (SSP), calculating the weighted contribution of the similar pixels (CWSP), calculating the conversion coefficient, and estimating the reflectance, can be executed along with the window moving operation. We will check that in the latter sections. For simplicity, PESTARFM means the PESTARFM based on the WPM, if it is not specified in the later sections.

## 3. Experimental Environment

In this section, we introduce the experimental environment. The environment consisted of ten physical machines. We used six machines to set up an OpenStack cloud platform [28]. OpenStack Ussuri, released in May 2020, was used. The six machines were Dell PowerEdge M610 servers, with each having two CPUs (24 cores in total), 48GB memory, and 10TB disk. We used one of them to provide storage for the virtual instances. All of the machines adopted the Ubuntu 16.04.4 LTS operating system with the 4.13.0-36-generic Linux kernel. We used a H3C Quidway S1026T switch to connect the six machines. All of the required packages that supported the private OpenStack cloud were installed with their native package. In order to simulate the elastic computing paradigm, we used the other four machines to build a small K8s cluster. In the small cluster, one serves as the K8s master note, and the others serve as workers. The K8s master node is a commodity PC with cores (Intel(R) Core (TM) i5-2400 CPU @ 3.10GHz), 16GB memory, and a 4TB disk. The three K8s worker nodes were each configured with a 12 core CPU (Intel(R) Core (TM) i7-8700 CPU @ 3.20GHz), 64 GB memory, and a 4TB disk. Table 2 lists all of the software required to build the K8s cluster in the following experiments. The worker nodes had the same Linux distribution as the master node. The physical networking of the K8s cluster was supported by the Cisco CVR100W Wireless-N VPN Router (100Mbps LAN). The H3C switch connected to the Cisco router.

Because our earlier work demonstrated the strength of Spark to process a massive volume of RS data, the primary goal of this study was to investigate the efficiency of the PESTARFM. All five RS images had 24,886 × 23,655 pixels (each pixel takes four bytes). In order to save some space, we will not introduce the images in the following experiments. We first set up a K8s cluster on top of the four nodes. Because K8s can run on almost all cloud platforms, we used the available computing nodes to simulate a resource-limited computing environment. We built a private local repository on the master node to store the relevant Docker images. We used an NFS client provisioner to ingress the external storage for the containers. We used Flannel to fabricate the container network. The Docker version was 18.06.1-ce, which was installed on every host. We used the Apache Ambari Blueprint (latest version 2.7.5) to construct a Hadoop cluster to store the RS images. Specifically, the containerized Ambari provided shell commands to easily create a containerized Hadoop cluster. All of the RS images were of Geotiff format, and were stored in the Hadoop cluster. The Spark operator consisted of the Spark AM, controller, and operator, and was installed by using Helm charts (see: https://github.com/GoogleCloudPlatform/spark-on-k8s-operator (accessed on 23 March 2021)). In order to simulate an elastic K8s computing cluster, we used the OpenStack cloud to add on-demand computing resources. As demonstrated in our earlier work, we can use an auto-scaling group (ASG) to easily add VMs. More specifically, when a VM boots up, the init daemon of the VM instance will automatically join the K8s cluster by issuing a kubeadm join command with a predefined token to the endpoint of the K8s API server. We then manually boot a VM instance (2VCPUs, 16GB memory) that connects to the K8s master node. We used a Heat template to define the scaling policy, which makes the OpenStack cloud provision VM instances when the CPU utilization rate is above 80%, and lasts about 1 min. The minimum, the desired, and the maximum of VM instances were set to 1, 5, and 10, respectively. The flavor of these VM instances was the same as the manually booted VM instance. There are many integrated solutions for managing different cloud resources [29]; for instance, Apache Libcloud is a standard Python library that abstracts away differences among multiple cloud provider APIs. Note that the OS and software used by VM images are the same as shown in Table 2. In order to dynamically change the specification of the Spark-operator containers, we used Flux to update the configuration of the deployed applications in the K8s cluster (https://github.com/fluxcd/flux2 (accessed on 23 March 2021)). Flux uses the kubectl command line interface to change the versions in the CRDs of applications. We used a script to update the version of the Spark-operator image in a short period (10 s in the experiment). Note that we did not change the specification of the Spark driver pods; we only changed the specification of the Spark executor pods without disturbing the execution of the Spark algorithms. By using the approach, we can easily change the replications of the Spark executor pods and the memory space size. Although this approach depends on the external tool to change the computing resources, the approach reduces the cost concerning the rewritten Spark AM. We set the maximum memory space for each SEP to 10 GB in order to consider the K8s master node resource constraints.

## 4. Results

### 4.1. Total Time Cost of PESTARFM on the Framework

For convenience, we abbreviate the size of the Spark driver memory, the size of the Spark executor memory, the memory of Spark executor instances, the partition number, and the time cost of the algorithm to DM, EM, N, P, and T, respectively. Note that the parameter is the internal parameter of the Spark-based fusion algorithm that determines the number of RDD blocks. Typically, the larger the parameter is, the smaller the tasks would be. The TSM was disabled in the first experiment. Note that the volume of the five RS images is about 10 GB. A Spark driver core was used. We used the K8s cluster to execute the containerized Spark application ten times, and calculated the average completion time of these tests. Before each execution, we manually removed all computing resources and reconstructed the K8s clusters according to the approach introduced in the above section. As shown in Table 3, the initial parameters of the DM, EM, and P were set to 8096 MB, 10,240 MB, and 20, respectively. PESTARFM takes approximately 913 s to complete when five Spark executors are used. In the experiment, we used the “kubectl scale deployment” command with the replication parameter set to the range from 5 to 25 for these tests. PESTARFM takes 910, 889, 869, 851, 839, 823, and 788 s respectively to complete when the maximum replicates of the Spark executors are set to 6, 7, 8, 10, 15, 20 and 25. The result shows that the time costs of PESTARFM on the framework demonstrate a decreasing trend. We noticed that the performance of the PESTARFM is not linear with the number of Spark executors. We suspect that computational tasks flock together on some computing hosts. We also found that the number of VM instances scales out to the desired numbers. Their average workload is low (about 11.7%) when using the Ganglia cluster tool. The results indicate that optimization strategies should be holistically considered in relevant aspects such as virtual networking and task scheduling.

### 4.2. Performance of the Task Scheduling Mechanism

In this section, we investigate the performance of the TSM in the second experiment. Typically, in Openstack clouds, Spark applications can leverage the auto-scaling group (ASG) to improve their efficiency. We compare the efficiency of PESTARFM using only ASG, with the efficiency of PESTARFM using ASG and TSM, to quantify the performance of the TSM. Typically, the partition number P determines the task size of Spark executors; we set the parameter P to 40 in the experiment in order to partition the data into smaller RDD blocks. As mentioned earlier, users can express the task scheduling expectation by using the weights. We set all of the weights to 0.33 in the experiment. We used the same operation in the former section to remove and construct the computing resources. The parameters DM and EM were set to 8096 MB and 2048 MB. As shown in Figure 4, the average time cost of the PESTARFM was reduced. By constraining the maximum replicates of the Spark executors to 10, the average time cost for the fusion of the five RS images was about 332 s. The results indicate that containerized Spark algorithms are suitable for executing small tasks rather than big tasks, which testify to our earlier conclusions. Typically, the ASG with TSM improves the performance of the PESTARFM. In the test that configured the maximum number of Spark executors to 100, the combination completed the RS image fusion in about 126 s. On average, the combination of ASG and TSM improves the efficiency of PESTARFM to 8.1% compared to the standalone ASG. From the tests, the maximum performance gain can be 13.6% ((146 − 126)/146) in the experiment.

We conducted the third experiment in order to investigate the impact of different weights of the TSM on the efficiency of PESTARFM. As discussed earlier, the weight W_1_ expresses the expectation to the TSM to dispatch tasks to newer computing hosts, and the weight W_2_ expresses the expectation to the TSM to dispatch tasks to the least overwhelmed computing host. The last W_3_ can be considered as the expectation to choose a computing host that has the maximum probability to finish tasks in a given period. We use (W_1_, W_2_, W_3_) to represent the weight set. Because we cannot elaborate all of the subsets, we chose the following four weight sets as (0.8, 0.1, 0.1), (0.1, 0.8, 0.1), (0.4, 0.4, 0.2) and (0.1, 0.1, 0.8). The period for the statistical calculation of the probability of the Spark executor pods was set to 10s in the experiment. The Spark parameters were the same as were used in the second experiment. As shown in Figure 5, all of the weight sets showed enhanced performance for the PESTARFM. There were a few differences among these sets. The red dashed line in the figure shows that selecting newer nodes rather than the least overwhelmed computing nodes would be more suitable for the fusion of big RS images. We observe that the more quickly the TSM dispatches tasks to newer nodes, the quicker the OpenStack cloud provisions VM instances. According to the OpenStack Dashboard, all of the 10 VM instances boot up in a minute during the execution period of PESTARFM. The performance of PESTARFM varies obviously (see: the orange line). When the experiment uses over 50 SEPs, the time cost of PESTARFM changes dramatically. The reason may be that tasks were assigned unevenly to the computing hosts. In a short period, the aforementioned probabilities of the overwhelmed hosts would be the same. Furthermore, Spark is typically a data locality computing paradigm that may also cause unbalanced processing. Nevertheless, the emphasis on both the newer and less overwhelmed computing hosts is a benefit for the improvement of the performance of PESTARFM (see: (0.4, 0.4, 0.2)). Comparing with the PESTARFM using only ASG, the performance gains of the PESTARFM using ASG with the TSM can be improved by 11.7%.

### 4.3. Performance of PESTARFM Using Containerized Hadoop

As it is known, the Hadoop yet another resource negotiator (YARN) has been widely used for cluster resource management, and can also support on-demand containers for BDPFs. We can use a containerized Ambari shell to build a containerized Hadoop cluster. In this section, we investigate the performance of PESTARFM using the containerized Hadoop cluster, and compare the performance with that using the proposed framework. For a fair performance comparison, we removed all OpenStack cloud resources, disabled the TSM, and reserved the K8s cluster before the fourth experiment. We first constructed a containerized Hadoop cluster and uploaded all RS images to the Hadoop cluster. In order to set the maximum memory for the YARN containers to 60 GB, and to restrict the maximum memory of each YARN container to 10 GB, we built a Docker image that used the Hadoop-3.1.2.tar.gz and jdk-8u202-linux-x64 tar.gz. We also packaged the RS fusion class to the Docker image. We used the containerized Ambari to construct the Hadoop cluster with the Docker image. As shown in Figure 6, the Hadoop cluster ran on the K8s cluster, which had 192 GB memory and 32 virtual cores.

The Spark driver memory size and Spark executor size were the same as used in the second experiment. Because Hadoop has no such mechanism to scale containers for Spark-based algorithms, we manually ran the following tests to execute the PESTARFM and calculated the averaged time cost of the algorithm. As shown in Figure 7, we attempted to use three types of Spark configurations for the PESTARFM. The first type changed the number of Spark driver cores (a); the second type changed the Spark executor cores (b). The last type changes the number of Spark executors (c). The time cost of PESTARFM was 585s when we used 1 CPU core for the Spark drivers. The time cost of PESTARFM was 417s when we used 4 CPU cores for the Spark drivers. We found that the PESTARFM always failed when a Spark driver requested 5 CPU cores. This is because the K8s master node has only four CPU cores. Note that we used 30 Spark executors in the experiment. The results were similar when the experiments used the second and third configurations. The shortest execution time of the PESTARFM using the containerized Hadoop cluster was 405s. The results show that the PESTARFM using the container Hadoop is slower than that using the proposed elastic computing framework.

## 5. Discussion

High-performance computing (HPC) for RS big data on cloud platforms is still a hot topic in the Geoscience community [30]. State-of-art researchers direct optimized task scheduling algorithms and auto-scaling cloud architectures to accelerate RS big data processing (RSBP). Take two representative works as an example. Sun et al. introduced a framework based on Spark for RSBP in CC [5]. Their contribution was that a task scheduling strategy was incorporated into the Apache Spark engine in order to minimize the total completion time of RS applications. Through a quantum-inspired evolutionary algorithm used to determine the optimal number of data partitions that a particular task can be divided into and processed in parallel on Spark, the framework showed a significant speedup for RSBD. However, their work did not consider the nature of the task failure of Spark applications that frequently occurs in heterogeneous cloud computing environments, as mentioned by themselves. Their experiment did not use the elastic computing paradigm that is the essential characteristic of CC. Our framework adopts the microservice architecture in a heterogeneous cloud with a commodity PC cluster environment. A widely-known cloud platform for RSBP is pipsCloud [31]. The authors introduced the pisCloud and suggested the use of the VM-based HPC clusters as services and RS generic parallel programming skeletons to implement message-passing interface (MPI)-based RS algorithms. As mentioned earlier, VMs are typically not so efficient as containers. Parallel RS algorithms based on the MPI model are still hard to implement, given that the detail of the underlying resource should be considered. As far as we know, there is only one work similar to our work in the community. Kang et al. suggested using the ASG of the OpenStack cloud platform to process legacy RS algorithms in VMs [32]. Our earlier work not only used the ASG in the IaaS layer of the OpenStack cloud platform but also the horizontal pod auto-scaler with K8s operators in the PaaS layer [33]. While our earlier work handled big vector data processing, this study focused on RSBP. Moreover, we incorporate containerized Spark operators to adapt the workload of computing hosts and change the capacities of Spark executors at runtime without disturbing the Spark algorithms. We named this new model the Spark-on-Kubernetes operator model in this study. Additionally, we proposed a PESTARFM for the fusion of RS big data. We are the first to explore Spark SQL to implement a parallel RS image fusion algorithm in this study. We proposed a task-scheduling mechanism that improves the performance of PESTARFM using the ASG only to about 11.7%. We also conducted several experiments to randomly change the memory size of SEPs, and the performance was greatly enhanced.

There are too many parameters that influence the performance of the PESTARFM. Recently, the authors conducted a comprehensive survey on automatic parameter tuning for big data processing systems [34]. According to their analysis, the number of key performance-aware configuration parameters for Spark applications is 16. Our proposed framework requires only the most important parameters, such as the Spark driver memory, the size of the Spark executor memory, and the memory of Spark executor instances. Benefiting from the containerization technology, when Spark executor pods launch on new computing hosts, their computing specifications are changed instantaneously. According to the experimental results shown in Figure 5, PESTARFM shows a similar performance under different parameter configurations.

## 6. Conclusions

In this study, we proposed a lightweight cloud-native framework of RS big data processing. The framework incorporated the Spark-on-K8s operator to improve the efficiency of a parallel RS image fusion algorithm. In the cloud-native computing layer, we adopted a container-based microservice architecture in order to easily update these loss-coupled service components by using K8s. Not only the IaaS scaling mechanism but also the PaaS scaling policy were used in the framework. Moreover, the parallel RS image fusion algorithms, PESTARFM, using Spark SQL was quite efficient for the fusion of RS big images according to our experiment. Additionally, we proposed a task scheduling mechanism to dispatch Spark executor pods to newer and less-overwhelmed computing hosts, thus reducing the time cost of PESTARFM in the framework. According to the experiments, the fusion of five big RS images completes in less than 4 min, while traditional ESTARFM uses multiple hours or even days. We believe that the computing architecture and the parallel RS image fusion algorithm are useful for the Geoscience community. Our future work will investigate factors such as network virtualization technologies, cloud object storage, and other scaling methods to boost the performance of cloud-native applications. Additionally, we will explore other science cloud computing environments to enhance the usability of the proposed framework.

## Figures and Tables

**Figure 1 sensors-21-02971-f001:**
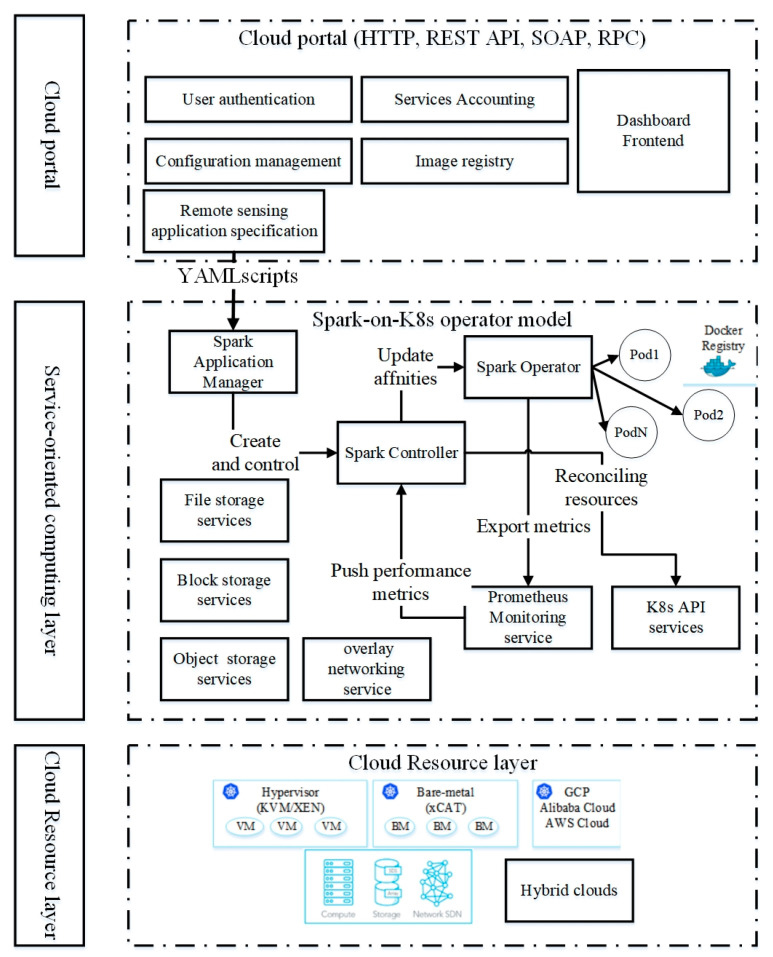
The lightweight cloud-native framework of RS big data processing.

**Figure 2 sensors-21-02971-f002:**
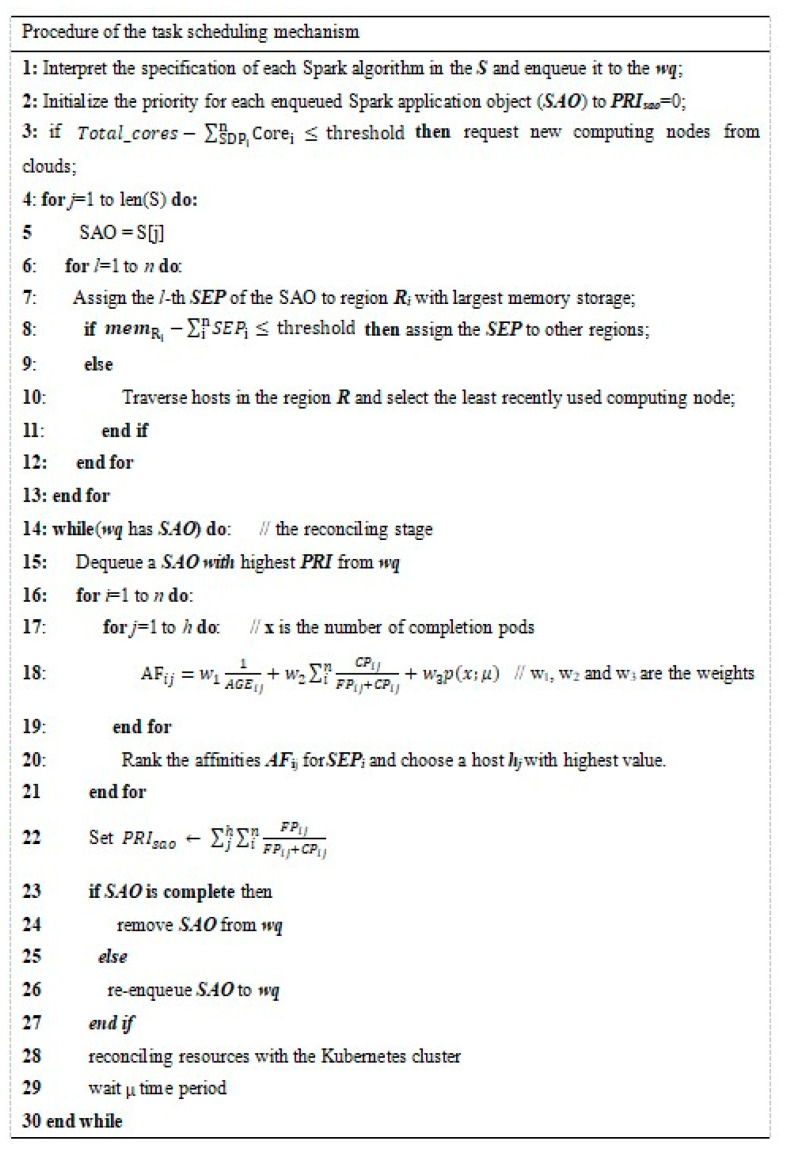
Pseudocode of the task-scheduling mechanism.

**Figure 3 sensors-21-02971-f003:**
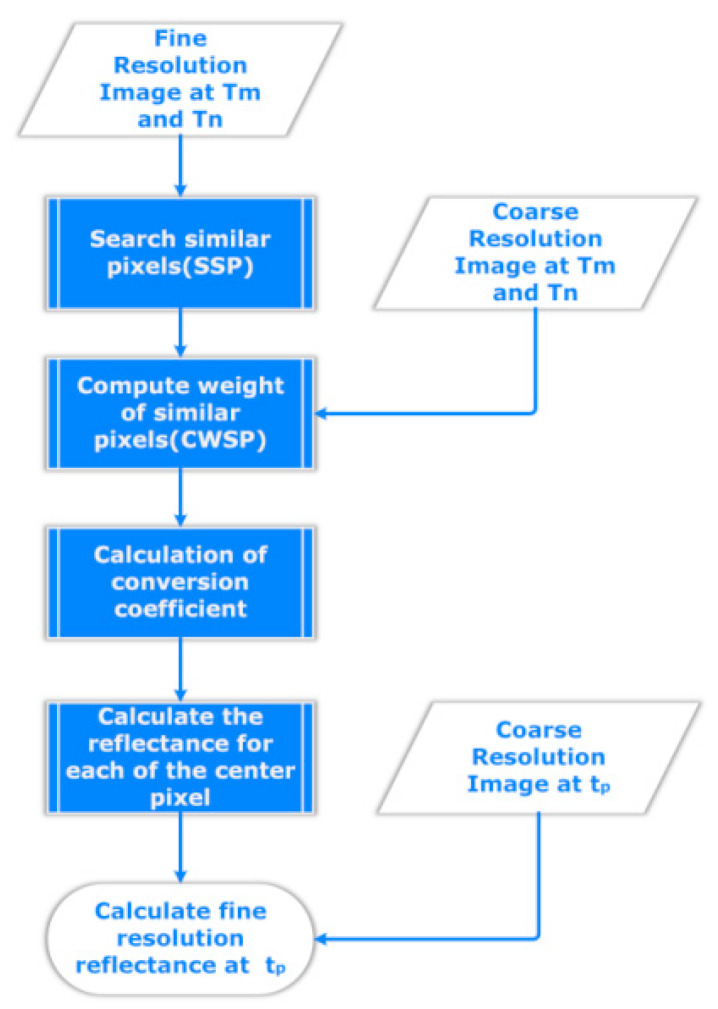
Flow chart of ESTARFM (enhanced spatial and temporal adaptive reflectance fusion model).

**Figure 4 sensors-21-02971-f004:**
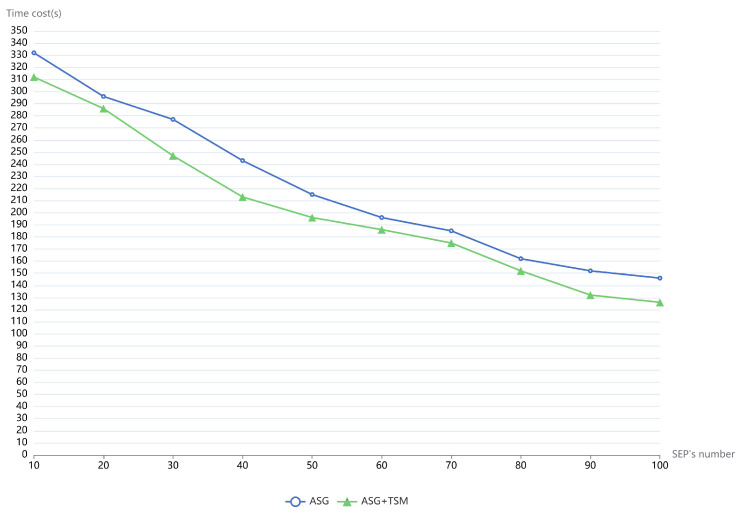
The time cost of PESTARFM using two models with the increasing number of Spark executors (one is using ASG, and the other is using the combination of ASG and TSM).

**Figure 5 sensors-21-02971-f005:**
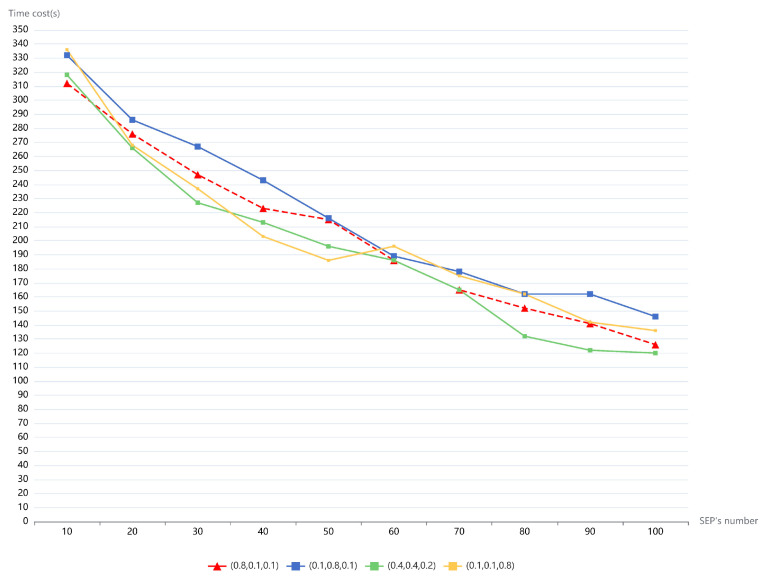
Time cost of PESTARFM using ASG, with TSM set with different weights.

**Figure 6 sensors-21-02971-f006:**
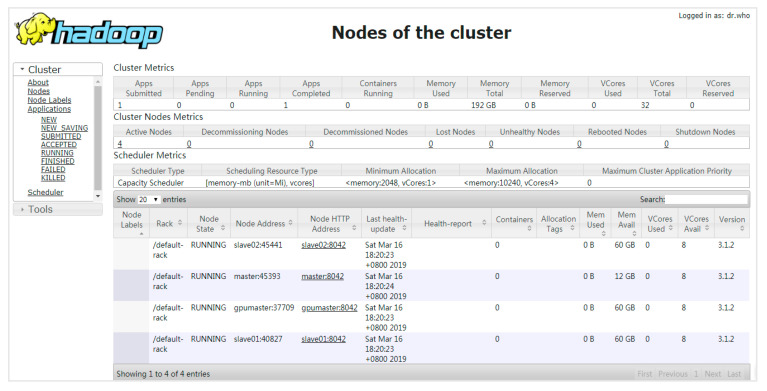
A containerized Hadoop cluster runs on the K8s cluster.

**Figure 7 sensors-21-02971-f007:**
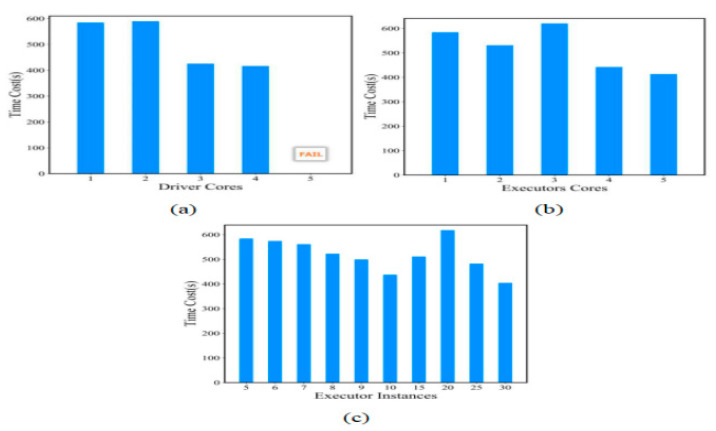
Time cost of PESTARFM using a containerized Hadoop cluster with three types of Spark parameters (Type (**a**) changing Spark driver cores; Type (**b**) changing Spark executor cores; Type (**c**) changing the number of Spark executors).

**Table 1 sensors-21-02971-t001:** Specifications of Sentinel-2A/B and Planet.

Sentinel-2A/B			PlanetScope		
Band Name	Band Width (μm)	Spatial Resolution (m)	Band Name	Band Width (μm)	Spatial Resolution (m)
Blue	460~525	10	Blue	450~515	3
Green	525~605	10	Green	525~600	3
Red	650~680	10	Red	630~680	3
NIR	785~900	10	NIR	845~885	3

**Table 2 sensors-21-02971-t002:** Software used for the K8s cluster.

Software	Version
Kubernetes	v1.13.4
Docker	18.06.1-ce
Spark	2.4.0
JDK	1.8.0_172
Etcd	v3.1.10
Flannel	v0.10.0-amd64
Kube-dns	1.14.7
Spark-operator	0.1.9
Scala	2.11.2
Hadoop	3.1.2
Spark	2.4.0

**Table 3 sensors-21-02971-t003:** Time cost of PESTRAFM on the framework (without TSM).

Driver Memory (MB)	Executor Memory (MB)	Executors	Partitions	Time Cost (s)
8096	10,240	5	20	913
8096	10,240	6	20	910
8096	10,240	7	20	889
8096	10,240	8	20	869
8096	10,240	10	20	851
8096	10,240	15	20	839
8096	10,240	20	20	823
8096	10,240	25	20	788

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
