# Peer review of "On-the-Fly Fusion of Remotely-Sensed Big Data Using an Elastic Computing Paradigm with a Containerized Spark Engine on Kubernetes"

_sensors, 2021, doi:10.3390/s21092971_

Round 1

Reviewer 1 Report

The authors have presented an interesting work about remotely-sensed image fusion by using an elastic computing paradigm.

The authors must analyze some minor points.

  • A final result of the framework must be provided and not just only its computational performance. For instance, the start argument in the abstract is "...image fusion is indispensable for analyzing long term Earth observation data", however no results showing this were presented.
  • enhance the algorithms and graphics quality (they are in low resolution)
  • Place the link on page 10 in a reference.
  • Considering the references, it seems that the paper was first submitted to the MDPI remote sensing journal. I suggest to add some references from sensors. (1 - Sensors Fusion and Multidimensional Point Cloud Analysis for Electrical Power System Inspection ; A Heterogeneous Edge-Fog Environment Supporting Digital Twins for Remote Inspections)
  • Finally, there are other frameworks for IOT big data processing (ML4IoT: A Framework to Orchestrate Machine Learning Workflows on Internet of Things Data) you should explore more this topic.

Finally, congratulations for the interesting work.

Reviewer 2 Report

The article proposed a lightweight cloud-native framework for the elastic processing of RSBD in this study. Both the scaling mechanisms provided by the Infrastructure as a Service and Platform as a Services of CC are used by the Spark-on-Kubernetes operator model with containerization virtualization technology. The model is used to construct the Microservice oriented computing para- 15 digm in CC. The study proposed a task scheduling mechanism (TSM) to dynamically change the Spark executor pods affinities to the computing hosts. It was proven that the mechanism can improve the performance of the PESTARFM using only PaaS scaling to about 11.7%.

This article presents the on-going fusion of Big Data remotely using a paradigm of elastic clack with a containerized spark engine on Kubernetes.
For the analysis of Earth observation data, the fusion of images detected from a distance is considered indispensable. Cloud computing provides the big picture for RSBD, but efficient merging on cloud computing platforms is not well solved. A lightweight native framework for cloud and elastic RSBD processing is proposed. The Spark-on-Kubernetes operator model will use the scaling mechanisms provided by IaaS and PaaS of cloud computing with container virtualization technology, a model used to build the computing paradigm oriented towards microservice in cloud computing, being implemented as a parallel model. Spatial and temporal adaptive reflective fusion.
The presented article has a significant impact on the presented field of work, especially in Big Data analysis. The article respects a scientific paper's ideal format; it is well structured, with suggestive images, tables, and clear graphics. However, I believe that the article is intended more for people with pre-existing studies in the field who already have a knowledge base.
The article provides plenty of useful and interesting information for people interested in innovative solutions regarding this specific field(Cloud computing, Big Data). This article is written in a scientific paper style, but the information can be a little difficult for novice readers to be understood.
Fix the minor grammar mistakes (row 13: “Infrastrure”, row 19, replace the “,” with a “.” etc.)
The graphs from Figure 7 should be a little bigger, as they are unreadable.
The conclusions should envision future work.
More related work regarding time critical cloud computing for fusion methods should be added, for example:
- Jia, Duo, et al. "A Hybrid Deep Learning-Based Spatiotemporal Fusion Method for Combining Satellite Images with Different Resolutions." Remote Sensing 13.4 (2021): 645.
- Štefanič, Polona, et al. "SWITCH workbench: A novel approach for the development and deployment of time-critical microservice-based cloud-native applications." Future Generation Computer Systems 99 (2019): 197-212.
- Corradino, Claudia, et al. "Combining Radar and Optical Satellite Imagery with Machine Learning to Map Lava Flows at Mount Etna and Fogo Island." Energies 14.1 (2021): 197.

Reviewer 3 Report

The authors proposed a lightweight cloud-native framework for the elastically processing of RSBD. The experiments show This framework can improve the efficiency of a parallel RS image fusion algorithm. The proposed idea is interesting, however, some revisions have to be made to claim the advantage of the proposed framework:

  1. The abstract section needs to be improved. The sentence in line 19 is incomplete.
  2. Could the authors explain what is the main contribution of the proposed framework in the Introduction?
  3. How much faster is the implementation of the proposed framework compared to other frameworks or platforms, and what are the advantages of the proposed framework?
  4. Figure 2 needs to be modified.
  5. What do the abscissa and ordinate represent in figures 4 and 5?
  6. Please explain why the partition number P and weights are set to 40 and 0.33.
  7. In Section 5, the discussion needs to be improved and the discussion should be about parameters.
  8. The English and format of this manuscript should be checked carefully.
